# Quantitative and temporal analysis of autophagy: Differential Response to amino acid and glucose starvation

**Katie R. Martin**[1�], **Stephanie L. Celano**[1☉], **Jessica D. Guillaume**[1], **Ryan D. Sheldon**[2], **Russell G. Jones**[3], **Jeffrey P. MacKeigan**[1]*

**1** Department of Pediatrics and Human Development, College of Human Medicine, Michigan State University, Grand Rapids, Michigan, United States of America, **2** Mass Spectrometry Core, Van Andel Research Institute, Grand Rapids, Michigan, United States of America, **3** Department of Metabolism and Nutritional Programming, Van Andel Research Institute, Grand Rapids, Michigan, United States of America

☉ These authors contributed equally to this work.
* mackeig1@msu.edu

## Abstract

Autophagy is a highly conserved, intracellular recycling process by which cytoplasmic contents are degraded in the lysosome. This process occurs at a low level constitutively; however, it is induced robustly in response to stressors, in particular, starvation of critical nutrients such as amino acids and glucose. That said, the relative contribution of these inputs is ambiguous, and many starvation medias are poorly defined or devoid of multiple nutrients. Here, we set out to create a quantitative dataset of autophagy across multiple stages in single, living cells, measured under normal growth conditions and during nutrient starvation of amino acids or glucose. We found that autophagy is induced by starvation of amino acids, but not glucose, in U2OS cells, and that MTORC1-mediated ULK1 regulation and autophagy are tightly linked to amino acid levels. While autophagy is engaged immediately during amino acid starvation, a heightened response occurs during a period marked by transcriptional upregulation of autophagy genes during sustained starvation. Finally, we demonstrated that cells immediately return to their initial, low-autophagy state when nutrients are restored, highlighting the dynamic relationship between autophagy and environmental conditions.

## Introduction

Macroautophagy/autophagy is a highly conserved pathway that recycles cytoplasmic contents via a vesicular trafficking pathway that culminates in the lysosome. Under normal conditions, autophagy serves a housekeeping function in cells by degrading long-lived proteins and clearing damaged organelles. However, during stress, autophagy is activated to liberate internal nutrient pools to support metabolism and

**Data availability statement:** The authors confirm that the data supporting the findings of this study are available within the article, its supplementary materials, and openly available in figshare (doi: https://doi.org/10.6084/m9.figshare.28797569).

**Funding:** This study was supported by grants and funding from the National Institutes of Health to J.P.M. (R01CA197398), and J.P.M. and R.G.J. (R01CA297993). J.P.M. funding from the National Cancer Institute (R21CA270588, R21CA252430, and R21CA263133) also supported this work. The funders had no role in study design, data collection and analysis, decision to publish, or preparation of the manuscript.

**Competing interests:** We have read the journal's policy and the authors of this manuscript have the following competing interests: R.G.J. is a scientific advisor for Agios Pharmaceuticals and Servier Pharmaceuticals and is a member of the Scientific Advisory Board of Immunomet Therapeutics. R.D.S. is a consultant with Theriome. J.P.M. has consulting agreements with Merck and scholarly activity with the Translational Genomics Research Institute and the Van Andel Research Institute. This does not alter our adherence to PLOS ONE policies on sharing data and materials.

circumvent cell death. There is a growing appreciation for the role of this stress response pathway in cancer, where autophagy has been shown to promote the progression of KRAS and BRAF-driven tumors by supporting tumor-intrinsic cancer cell survival, and also, by regulating additional cell types in the tumor microenvironment (i.e., immune cells) [1–9].

The gatekeeper of autophagy is ULK1, a serine/threonine kinase that triggers the nucleation of an isolation membrane, or phagophore, the earliest autophagic membrane [10–12]. The synthesis of this cup-shaped structure is largely promoted by the class III PI3K (PIK3C3) and its lipid product, phosphatidylinositol-3-phosphate (PI(3)P). PI(3)P decorates autophagic membranes and recruits effectors, such as double FYVE domain-containing protein 1 (DFCP1) and WD repeat domain phosphoinositide interacting protein 1 and 2 (WIPI1 and WIPI2) [13–15]. VPS34 activity is required for the downstream activation of ATG9, a transmembrane protein that facilitates lipid transport to contribute to the expansion of the phagophore membrane [16–20]. Maturation of the phagophore and eventual closure into a complete autophagosome is executed by two ubiquitin-like conjugation systems. The first system involves covalent binding of autophagy related protein 12 (ATG12), a ubiquitin-like protein, to autophagy related protein 5 (ATG5) and subsequent incorporation into a large oligomer with autophagy related protein 16 (ATG16) at the phagophore [21]. The second system involves the classic autophagosome-marker, microtubule associated protein 1 light chain 3 beta (MAP1LC3B; henceforth, LC3B) (Atg8 in yeast), which becomes covalently attached to phosphatidylethanolamine on autophagic membranes [22,23]. The location and enabling of LC3 conjugation to the vesicle is controlled by ATG12-5-16, which functions as an E3-like enzyme [24]. The completion of this process is marked by the autophagosome's direct fusion with a lysosome (generating an autolysosome), or with an endosome destined for the lysosome (generating an amphisome), and the degradation of sequestered cargo.

Autophagy is tightly regulated by environmental conditions, most notably, it is activated by amino acid stress through regulation by mammalian target of rapamycin (MTOR; specifically MTOR complex 1 or MTORC1) (Fig 1A). Under amino acid sufficiency, activated MTORC1 restrains autophagy through inhibitory phosphorylation of the ULK1 complex (at Ser758) [10–12,25,26]. This serves to limit autophagy-mediated catabolism during times of sufficient exogenous nutrient supply. Upon amino acid withdrawal, MTORC1 is repressed, thereby relieving its inhibition of ULK1 and activating autophagy.

To date, the prevailing dogma has been that in addition to amino acid stress, autophagy can also be activated by energetic stress through regulation of the energy-sensing AMP-activated protein kinase (AMPK) (Fig 1A). In this model, during energetic stress (i.e., decreased cellular ATP/AMP ratio), AMPK is activated and inhibits MTORC1 by at least two mechanisms: the phosphorylation of TSC2, an upstream negative regulator of MTORC1, and phosphorylation of RAPTOR, a component of the MTORC1 complex [27,28]. This repression of MTORC1 should contribute to ULK1 activation and autophagy induction. In addition, it is known that AMPK also directly phosphorylates ULK1 at several sites [29–31]. While this regulation of AMPK

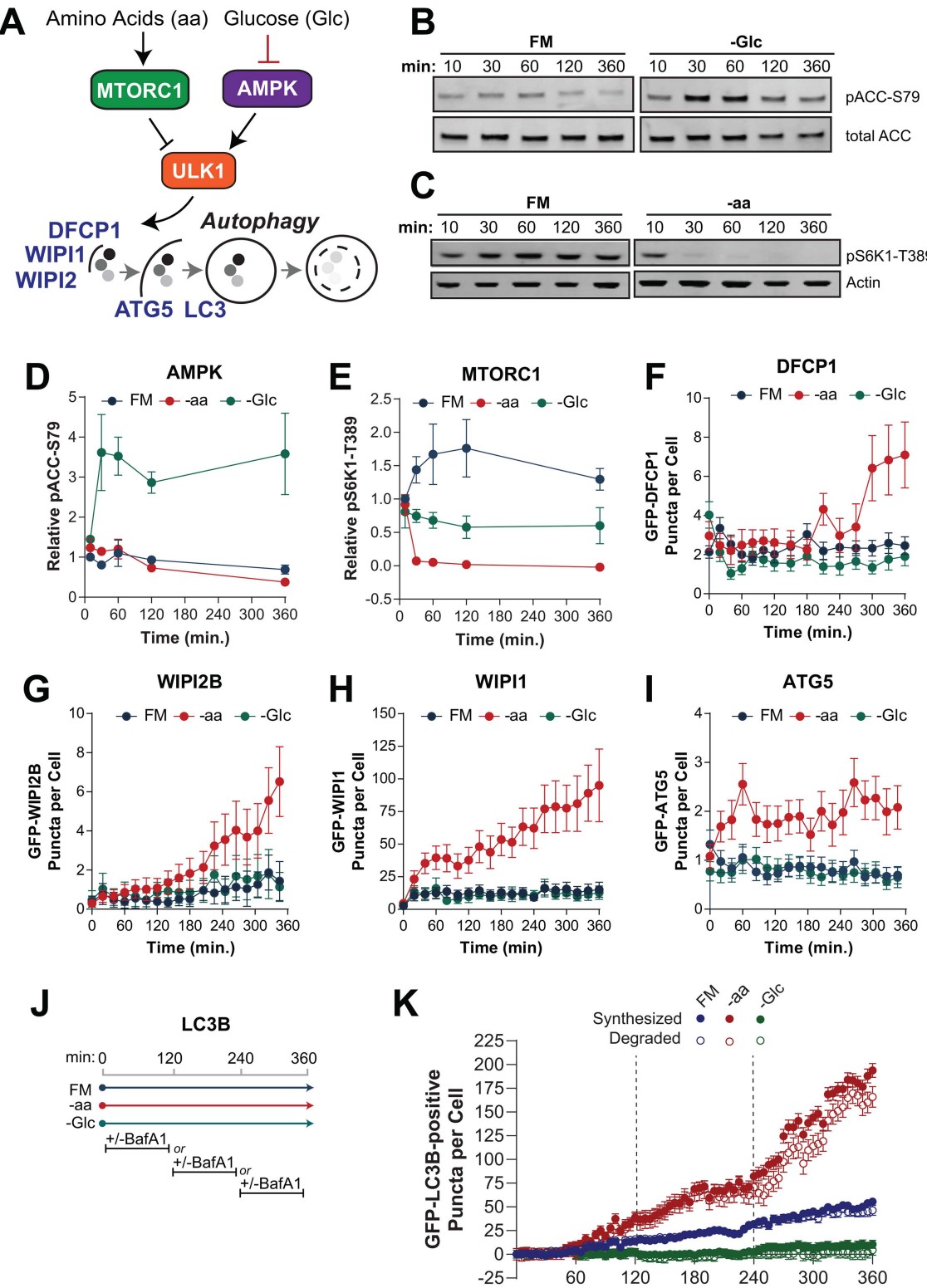

**Fig 1. Amino acid starvation, but not glucose starvation, induces autophagy in U2OS cells. (A)** MTORC1 and AMPK activity is influenced by amino acids and glucose, respectively, and regulate ULK1 activity, which induces autophagy. **(B-E)** U2OS cells were treated with full media (FM), FM lacking glucose (-Glc), or FM lacking amino acids (-aa) for the indicated times and cell lysates analyzed by immunoblot for ACC, total and

phosphorylated at S79 **(B, D)**, and S6K1 phosphorylated at T389 and actin as a loading control **(C, E)**. Immunoblots were quantified and signal normalized to total ACC (for D) or actin loading control (for E) plotted. Symbols represent mean of 3 independent experiments and bars are s.e.m. **(F-I)** Monoclonal U2OS cell lines expressing DFCP1 **(F)**, WIPI2B **(G)**, WIPI1 **(H)**, or ATG5 **(I)** were treated with FM (blue), -Glc (green), or -aa media (red) for 6 hours and subjected to live-cell fluorescent imaging. GFP-puncta (objects) for each reporter were quantified from single cells. Trajectories include mean objects per cell (symbols); bars represent 95% CI. **(J-K)** GFP-LC3B objects were quantified from cells treated with FM, -aa, or -Glc in the presence of BafA1 (to prevent lysosome degradation) or a vehicle control in 2 hour increments **(J)**. The number of GFP-LC3 puncta synthesized (solid symbols) and degraded (open symbols) from time 0 min was calculated and plotted. The dashed lines demarcate where individual datasets were collected and data stitched together. Trajectories include mean puncta per cell (symbols); bars represent standard deviation.

on ULK1 was long thought to be a positive input, that is, AMPK activation increases ULK1-dependent autophagy, this has been challenged in recent years [32–36]. For example, glucose starvation has been found to trigger AMPK-mediated phosphorylation of ULK1 at Ser556, which though initially viewed as an activating event, has recently been shown to inhibit autophagic flux [37–39]. Thus, there is now a growing appreciation that the relationship between AMPK and autophagy is complex and in fact, AMPK may contribute to both the activation or inhibition of autophagy depending on the severity or level of energetic stress [37,40].

A major source of inconsistency across autophagy studies stems from differences in the exact media formulations used. Starvation media are often poorly defined and can vary by amino acid and glucose content, the presence and type of fetal bovine serum (which, unless dialyzed, contains undefined amounts of nutrients, including amino acids and glucose), and other components (reviewed in [40]). In addition, the starvation duration can range dramatically, from minutes (short duration) to days (long duration), further complicating comparisons. Given that autophagy is a highly dynamic process and technically challenging to measure with precision, it is thus not surprising that these disparate experimental conditions yield conflicting conclusions.

To address these limitations and to provide clarity on how nutrient stress affects autophagy, we used a well-characterized cancer cell line, U2OS, and fluorescent sensors that mark distinct stages of the autophagy pathway. We monitored autophagy during defined amino acids or glucose starvation over the course of 6 hours, capturing measurements at 5–10 minute intervals to generate a high-resolution view of autophagic dynamics. For this, we formulated defined medias to precisely limit these nutrients in isolation while controlling all other media components. Using these medias, we found that withdrawal of amino acids, but not glucose, induced autophagy in U2OS cells. We confirmed that MTORC1 regulation of ULK1 and the magnitude of autophagy were tightly linked and correlated with amino acid availability. Moreover, we found that while autophagy was activated immediately in response to amino acid deprivation, further upregulation occurred several hours later, concomitant with transcriptional upregulation of autophagy machinery. Finally, we demonstrated that cells were primed to return to a low-autophagy state when nutrients were replenished. In addition to these well-supported conclusions, we provide single-cell data from living cells captured at high temporal resolution across multiple stages of autophagy as a resource for other researchers.

## Results

### Starvation of amino acids, but not glucose, induces robust autophagy in U2OS cells

To understand the relative contribution of the major physiologic inputs to autophagy, we measured multiple stages of the pathway in U2OS cells cultured in each of three defined medias (see S1 Table for full formulations): *1)* custom full media ("FM"), which contains the concentrations of amino acids, growth factors, glucose, and vitamins found in RPMI-1640 supplemented with 10% dialyzed fetal bovine serum (FBS); *2)* amino acid starvation media ("-aa"), which is FM without amino acids; and *3)* glucose starvation media ("-Glc"), which is FM without glucose. First, we confirmed that -Glc media increased AMPK activity as measured by phosphorylation of the AMPK substrate, ACC, at Ser79, an effect that was immediate and maintained throughout the entire 6 hour period (Fig 1B, D). We also established that amino acid starvation inhibited MTORC1, as indicated by a complete loss of phosphorylation of its substrate, S6K1, at T389 within 30 minutes

(Fig 1C, E). We also detected a decrease in pS6K1-T389 in glucose-free media (-Glc) relative to full media (FM), consistent with the known negative regulation of MTORC1 by AMPK through TSC2 and RAPTOR [27,28].

To determine the consequences of these medias on autophagy, we used a panel of monoclonally derived U2OS cell lines expressing EGFP-fusions of key autophagy machinery including DFCP1, WIPI1, WIPI2B, ATG5, and LC3B (Fig 1A). Cells were plated in FM and then switched to either FM, -aa, or -Glc before fluorescent live-cell imaging over 6 hours. We found that fed cells (cultured in FM) generally expressed few (2.1 +/- 1.4) DFCP1-positive puncta, which represent omegasomes, the crescent-shaped structure that supports the forming phagophore, and these were relatively stable through the 6 hour imaging period (Fig 1F, blue). Amino acid starvation (-aa) increased DFCP1-positive puncta, primarily after 3 hours of treatment, peaking at an average of 7.1 puncta per cell by 6 hours (Fig 1F, red). In contrast, glucose-free media (-Glc) failed to induce DFCP1 puncta formation (Fig 1F, green). WIPI2B-positive puncta were also lowly abundant and responded similarly to -aa, rising from 0.3 to 6.5 puncta per cell on average (>20-fold), most substantially in the final 3 hours of starvation, while failing to respond to -Glc as well (Fig 1G). The second PI(3)P effector we quantified, WIPI1, also responded exclusively to -aa; however, this marker showed an immediate increase in puncta, rising from approximately 5–40 puncta per cell in the first hour of -aa, followed by a slow and steady increase through the remainder of the starvation period (Fig 1H). Similar to WIPI1, the first hour of -aa triggered a doubling in the number of ATG5-positive puncta, although these structures were overall very rare (1–2 per cell) (Fig 1I).

As LC3B is known to be conjugated to the growing autophagic membrane and degraded in the lysosome along with cargo, measuring it requires a more complex experimental design. We adopted a straightforward kinetic experimental design to approximate both the synthesis and degradation of GFP-LC3B-positive autophagic vesicles (AVs), which we observe as GFP-positive puncta, in FM, -aa, and -Glc medias. Our calculations were based on the principle that the AVs in a cell reflect a dynamic pool continually influenced by the synthesis of new AVs and loss of existing AVs to lysosomal degradation. We distinguished synthesis and degradation by measuring AVs over time in both the presence and absence of bafilomycin A1 (BafA1),a proton pump inhibitor that prevents lysosomal degradation [41]. Because BafA1 prevents degradation, synthesized AVs are quantified as those that accumulate over time in the presence of BafA1. We then compare AV counts observed with and without BafA1 to approximate AV degradation (see *Methods* and Table S2). Because the duration of BafA1 treatment must be limited, we spiked in BafA1 for either the first, middle, or final two hours of the 6-hour treatment period and quantified LC3-positive vesicles (Fig 1J). Similar to the other markers, we found that -aa caused a significant increase in LC3-positive vesicle synthesis, an effect that was apparent immediately and grew more robust over time (Fig 1K). In contrast, not only did glucose-free media not induce AV synthesis, glucose-free media actually suppressed it below the basal levels observed in FM (Fig 1K). For all conditions, the rate of AV synthesis and degradation was well-balanced (Fig 1K), underscoring the ability of these cells to regulate autophagy efficiently. Taken together, these data strongly suggest that autophagy is upregulated in U2OS cells in response to starvation of amino acids, but not glucose withdrawal.

To understand why glucose starvation failed to induce autophagy in our experiments, we compared the state of two major phosphorylation sites on ULK1, serine 758 (S758) and serine 556 (S556). First, we found that glucose starvation failed to reduce pULK1-S758, a critical inhibitory phosphorylation site on ULK1 regulated by MTORC1 [30], whereas this site was dephosphorylated by aa-starvation (S3A-B Fig). Second, we found that glucose starvation triggered and sustained high levels of pULK1-S556, which again was in contrast to a time-dependent dephosphorylation observed with aa-starvation (S3C Fig). Emerging evidence has found S556 to be an AMPK-mediated phosphorylation site, which is consistent with the increased AMPK activity we detected under glucose-free media conditions, and that it is associated with inhibition of ULK1 and autophagic flux, which is also consistent with our data [38–40].

## MTORC1-mediated autophagy is tuned to amino acid levels

After observing that amino acids are the dominant driver of autophagy in U2OS cells, we wanted to more precisely establish the relationship between amino acid levels and autophagy induction. For this, we titrated amino acids from 100% (the

level of each amino acid in the RPMI-1640 formulation and our custom FM) down to 10%, 5%, and 0%, while keeping all other media components constant. At the end of the 6 hour treatment period, we saw that decreasing amino acids led to decreasing levels of pULK1-S758, with 0% aa causing a near complete loss of pULK1-S758, similar to that observed with pS6K1-T389 in Fig 1 (Fig 2A). These data were fit to a sigmoidal dose-response curve, which revealed an $EC_{50}$ of 6.4% aa (+/- 1.2% aa) (Fig 2B). We then measured LC3-positive AV synthesis by supplementing BafA1 into these medias and imaging them between 4 and 6 hours of treatment. We found that reduction of amino acids in the media increased the rate of LC3 synthesis, with an $EC_{50}$ of approximately 7.6% aa (+/- 2.5% aa) (Fig 2C-D). We confirmed that there was a linear negative relationship between pULK1-S758 and LC3 vesicle synthesis, consistent with relief of this phosphorylation site on ULK1 being a critical determinant of autophagy synthesis (Fig 2E). Moreover, this data suggests that amino acid levels must drop significantly from the amount in FM before there is an appreciable upregulation in autophagy.

## PI(3)P effectors are differentially regulated during starvation

Upon aa starvation, cells immediately increased the abundance of puncta marked by WIPI1, a PI(3)P effector, increasing nearly 8-fold within the first hour (Fig 3A). To determine whether these WIPI1-positive structures reflected newly synthe-sized PI(3)P-positive vesicles, we quantified puncta positive for EGFP-2xFYVE, a universal marker of PI(3)P-positive membranes in cells. We found that 2xFYVE-positive puncta were abundant in cells prior to starvation, with an average of 100 puncta per cell (Fig 3B). During the first hour of aa-starvation, this abundance did not change, although it increased modestly later in starvation (Fig 3B). This suggests that when cells are first aa-starved, cells respond by recruiting WIPI1 to pre-existing PI(3)P-positive membranes, potentially of endosomal origin as these are typically the most abundant EGFP-2xFYVE-positive structures [42]. In contrast to WIPI1, WIPI2B responded more slowly with robust increases in WIPI2B-positive puncta later in the aa starvation (Fig 3A). Specifically, it took 2–3 hours of aa starvation for 50% of the cell population to have at least 1 WIPI2B puncta (Fig 3C).

We next wanted to establish the requirement for PI(3)P both early and late in aa-starvation (when WIPI1 and WIPI2B respond most dramatically, respectively). For this, we treated cells with a small molecule inhibitor of VPS34 (compound 31), the lipid kinase that produces PI(3)P. Compound 31 significantly reduced PI(3)P, as evidenced by loss of EGFP-2xFYVE puncta (Fig 3D). Moreover, we observed an accompanying reduction in LC3-positive vesicle synthesis when measured both early (1 hour into aa-starvation) or late (5 hours into aa-starvation), consistent with these PI(3)P dynamics being critical for autophagy upstream of LC3 (Fig 3E).

## Delayed autophagy is associated with transcriptional upregulation of machinery

We were intrigued by the observation that autophagy was particularly high late in starvation, beginning 3–4 hours into aa-deprivation, as evidenced by the increased abundance of nearly all markers measured (Fig 1). Though amino acids were removed from the extracellular media at time 0, we wondered if the intracellular level of amino acids might shed light on the magnitude of autophagy, in particular, those known to regulate autophagy, including leucine, arginine, methionine, asparagine, and glutamine [43]. To address this, we employed liquid chromatography mass spectrometry (LC-MS) to quantify amino acids from cells 0 min, 20 min, 1 hour, or 4 hours into aa-starvation. We found that there was nearly com-plete depletion of these amino acids within 60 minutes of aa-starvation (Fig 4A) and no significant change in any amino acid between the 1 hour and 4 hour timepoints that might explain a change in autophagy (Fig 4B).

Next, we reasoned that this timeframe may also be consistent with the induction of gene expression and transcription of autophagy genes [44,45]. To determine whether a transcriptional program might underlie this increase in autophagy, we harvested RNA from cells after 1, 2, 4, or 6 hours of aa-starvation or after the same duration of treatment with FM. We then analyzed transcript expression by RNAseq. Indeed, we found a significant transcriptional response to aa-starvation (Fig 4C) with 1,642 and 1,766 genes increased or decreased at least two-fold, respectively (S3 Table). Among the genes upregulated during aa-starvation, we identified enrichment of pathways consistent with a stress response, for example,

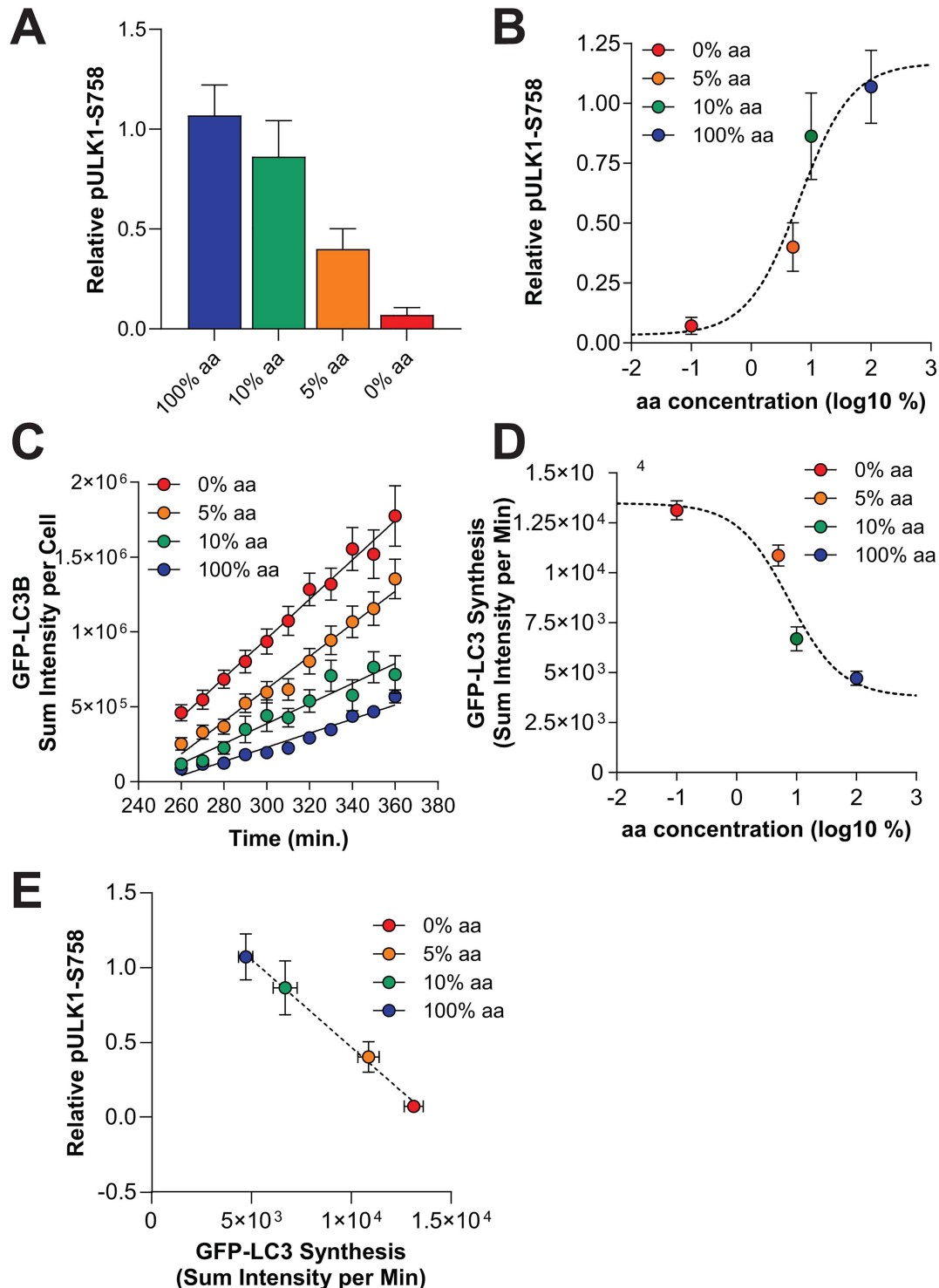

**Fig 2. ULK1 phosphorylation and autophagy levels are tightly associated and regulated by amino acid levels. (A-B)** U2OS cells were treated for 6 hours with FM (blue), indicated as 100% aa (the concentration found in RPMI-1640), or 10% (green), 5% (orange), or 0% (red) of that amino acid concentration. Cells were lysed and ULK1 phosphorylated at S758 quantified (relative to actin loading control and normalized to time 0 controls) **(A)**. Bars represent means of 3 biological replicates. The data in (A) was fit to a sigmoidal dose-response curve (dashed line) to generate an $EC_{50}$ of 6% aa **(B)**. **(C-D)** Cells were treated with the medias described in A and imaged live from hours 4-6 in the presence of BafA1 (as in Fig 1K). GFP-LC3 puncta were

quantified from cells and sum intensity plotted (this is the sum of the intensity of all GFP-positive pixels, an output used to avoid potential issues with aggregated vesicles). Trajectories include mean objects per cell (symbols); bars represent s.e.m.; black lines represent simple linear regression **(C)**. The GFP-LC3 synthesis rates from the linear regression lines in (C) across amino acid concentrations were fit to a sigmoidal dose-response curve (dashed line) to generate an EC$_{50}$ of 7% aa **(D)**. **(E)** The rate of GFP-LC3 synthesis (derived from linear regression analysis, shown in (C) and the relative level of pULK1-S758 (from A) plotted to show a negative, linear association (dashed line, $r^2 = 0.815$).

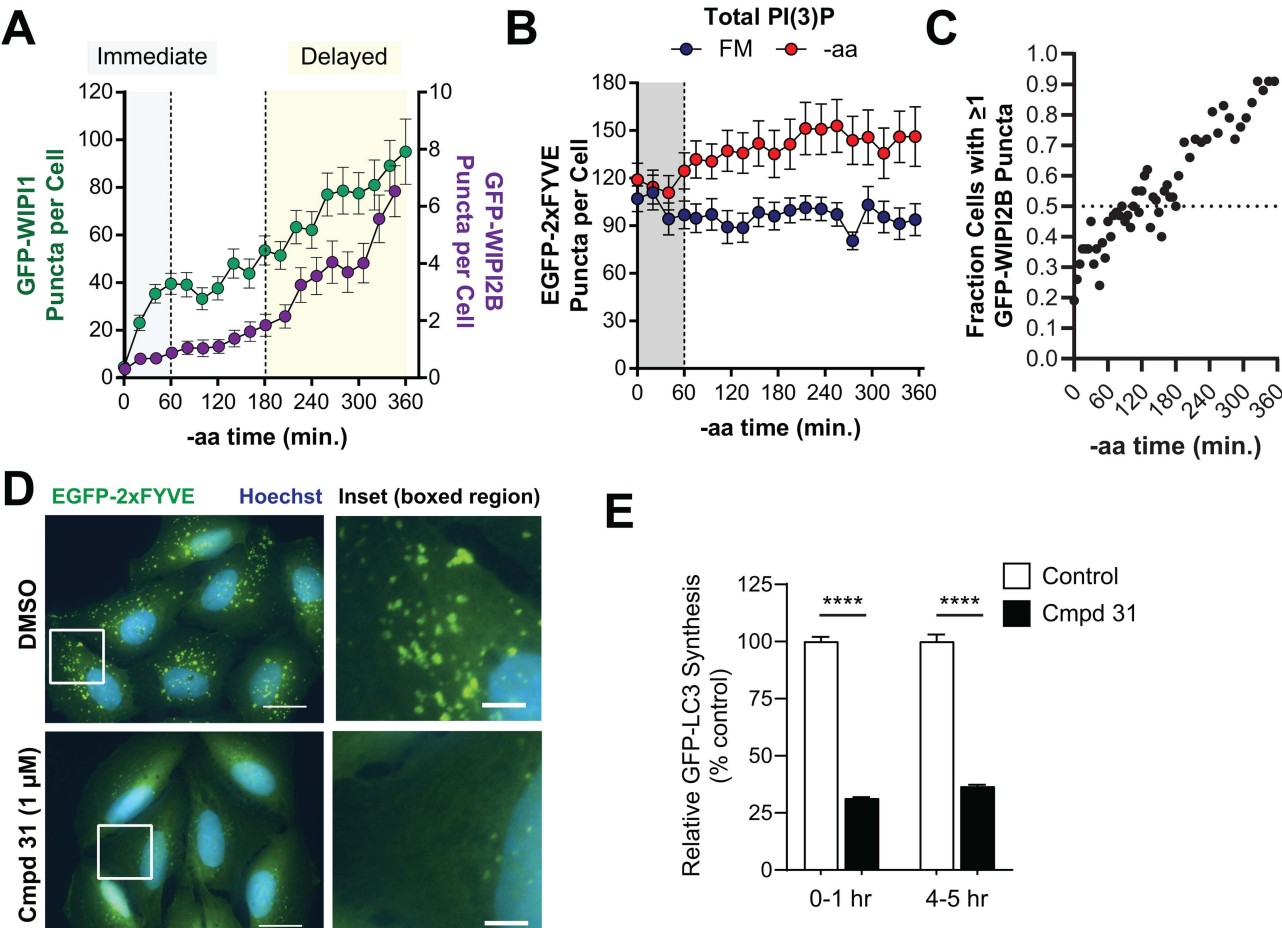

**Fig 3. PI(3)Peffectors, WIPI1 and WIPI2B, show distinct responses to aa starvation. (A)** GFP-WIPI1 (green, left Y-axis) and GFP-WIPI2B (purple, right Y-axis) object counts over the 6 hour -aa treatment were overlaid. The gray region indicates the immediate starvation period (0-1 hours), and the yellow highlights the period of delayed autophagy under sustained starvation (3-6 hours). **(B)** EGFP-2xFYVE puncta (PI(3)P-positive cell membranes) were quantified from cells under FM (blue) or -aa (red) treatment. Note a lack of substantial puncta increase in the immediate (0-1 hour) period (gray shading). **(C)** The fraction of cells containing at least 1 GFP-WIPI2B puncta is plotted with time of -aa starvation. The dashed line represents 50% of the cell population. **(D)** Representative EGFP-2xFYVE puncta in U2OS cells treated with a VPS34 inhibitor (1 µM compound 31, lower panels) or vehicle control (upper panels). Blue = Hoechst nuclear stain; green = EGFP-2xFYVE; captured with a 60x oil objective. Scale bars in left panels are 20 µm and scale bars in right panels (insets) are 5 µm. **(E)** GFP-LC3 synthesis with BafA1 in the presence of compound 31 (1 µM) or vehicle control. BafA1 was added for 1 hour during either the first hour of -aa starvation ("0-1 hr" bars) or after 4 hours of -aa starvation ("4-5 hr" bars). Data shown represent GFP-LC3 puncta synthesis relative to vehicle control. Symbols represent mean and bars are s.e.m. **** = adjusted $p < 0.0001$, one-way ANOVA.

the Reactome pathways "Response of EIF2AK1 (HRI) to heme deficiency" and "NGF-stimulated transcription", which involve the integrated stress response and immediate early gene induction, respectively (S4 Table) [46,47]. Moreover, among the most significantly upregulated genes were *ATF3*, *EGR1*, and *FOS,* immediate early genes known to be

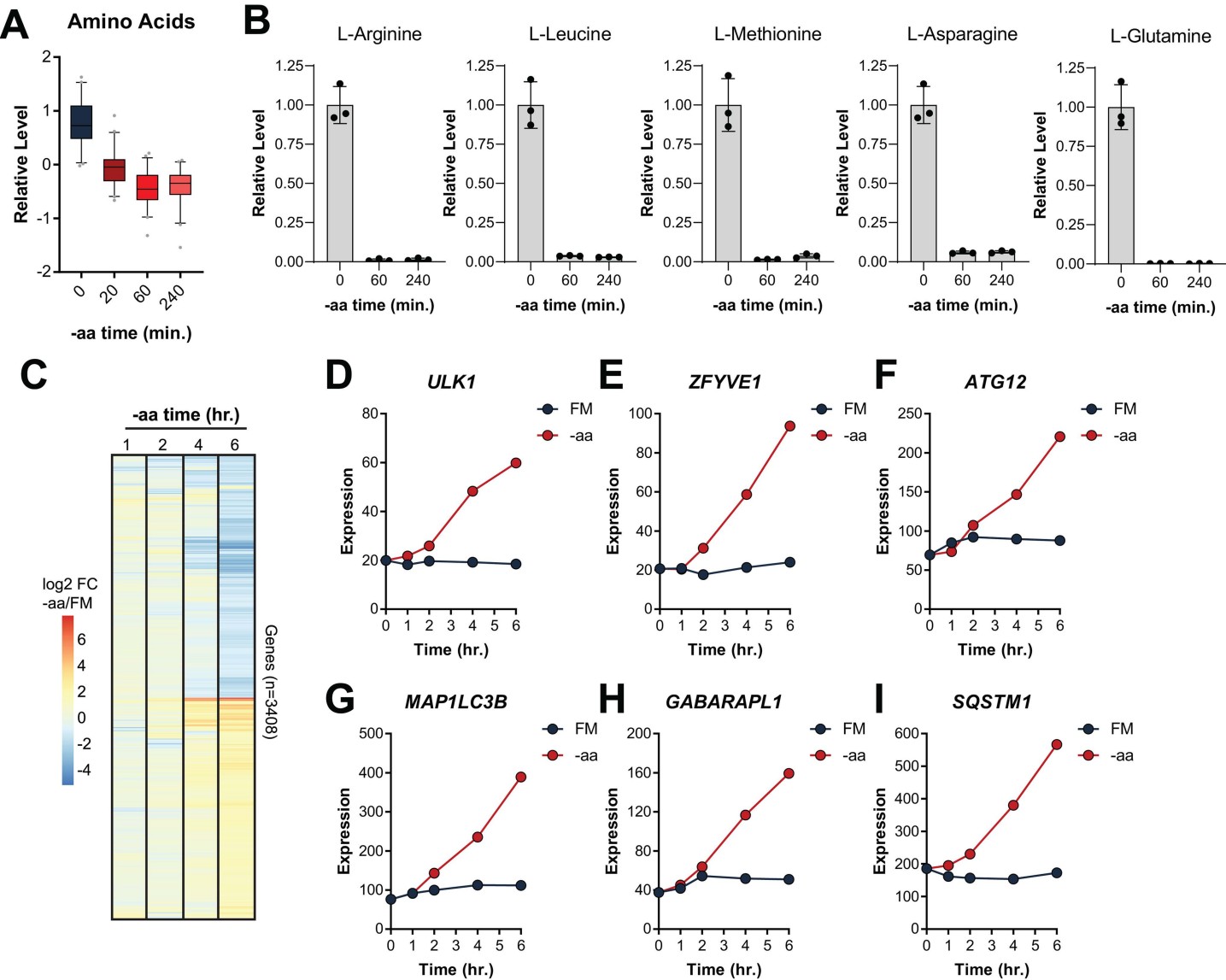

**Fig 4. Autophagy machinery is transcriptionally upregulated with sustained amino acid starvation. (A)** Intracellular amino acids were measured from triplicate samples after 0 min, 20 min, 1 hour, or 4 hours of -aa starvation using mass spectrometry. Levels shown relative to time 0 (non-starved cells). Boxes are 25th to 75th percentile; line in box is median; whiskers are 5th and 95th percentiles. Symbols are datapoints outside box and whiskers. Note, we detected 16 of the 20 amino acids provided in FM. **(B)** Intracellular amino acids known to regulate MTORC1 and autophagy are plotted individually. Bars are mean (relative to pool size at time 0) and s.e.m. from triplicate samples (individual replicates shown as symbols). **(C)** RNAseq profiling was performed from cells cultured during -aa starvation for the indicated times compared to FM treatments. Differentially expressed genes (log2 fold-change +/-1 -aa versus FM at 6 hours) are shown. **(D)** Expression of core autophagy genes (in counts per million, CPM) is shown at the indicated times of FM or -aa treatment.

induced as part of the amino acid response (AAR), a cytoprotective mechanism induced by amino acid withdrawal [48]. Importantly, autophagy genes (identified in the Human Autophagy Database [49]) were significantly enriched among those increased (two-sided p = 0.0009, Fisher's exact test). Specifically, we observed an upregulation of genes encoding core autophagy proteins, including *ULK1*, *ZFYVE1* (DFCP1), *ATG12*, *MAP1LC3B* (LC3B), *GABARAPL1* (an LC3-like molecule), and *SQSTM1* (p62, a selective autophagy cargo adaptor) (Fig 4D).

## Nutrient replenishment immediately restores cells to their basal autophagy level

After characterizing the response of cells to aa withdrawal, we wondered how quickly they would respond to the replenishment of amino acids following starvation. To test this, we starved cells of aa for 6 hours and then restored amino acid levels fully and quantified DFCP1, WIPI1, WIPI2B, or LC3B for 1 hour (Fig 5A). We found that all cells responded remarkably quickly to aa replenishment, with restoration of the basal state within 15–20 minutes (Fig 5B-G). This swift response led us to wonder whether the disappearance of autophagic vesicles upon nutrient replenishment was from degradation in the lysosome or potentially another route of vesicle disassembly. To explore this, we quantified LC3-positive AVs during nutrient replenishment in the presence of BafA1, which as described earlier, prevents lysosomal degradation. If the vesicles that disappeared had done so through lysosomal degradation, we reasoned that their rapid decrease should be negated with BafA1 treatment. In fact, this is what we observed, instead seeing an increase in AV abundance with BafA1 in aa-starved cells replenished with FM (S2A Fig). To explore this further, we calculated the number of LC3-positive vesicles (puncta) synthesized and degraded during nutrient replenishment (using paired measurements with and without BafA1) and stitched it onto FM or aa-starvation data (from Fig 1K) (S5 Table). We determined that once FM was replenished to aa-starved cells, synthesis immediately slowed while degradation proceeded at a higher rate (S2B–C Fig). Within 30–60 minutes, synthesis and degradation were well-balanced and maintained at a rate (slope) very similar to the FM control cells (S2B Fig).

Taken together, this nutrient replenishment data suggests that cells are more buffered to the induction of autophagy in the face of starvation, showing a variable and often later autophagy induction, whereas they are primed to return to their basal homeostatic rate of autophagy the moment a stressor, in this case aa-starvation, is relieved.

## Discussion

Here, we have provided a detailed and quantitative description of autophagy induced by amino acid deprivation in a widely used cancer cell line for signal transduction and autophagy research, including five protein phosphorylation sites and five fluorescent reporters for autophagy proteins. We supplemented these data with measurements of intracellular amino acids and mRNA transcript abundances. Collectively, we provide these data as a resource for the research community, for instance, to use in constructing or testing models or for exploring single cell behavior across a population. While we captured our data in a single cell line (U2OS), we expect our conclusions to have broad application to many cancer cell lines given the evolutionary conservation of the autophagy machinery and signaling networks involved.

Our most notable observation was that U2OS cells strongly induce autophagy in response to loss of amino acids, but fail to do so in response to glucose starvation. In fact, we found reduced LC3 vesicle dynamics during glucose starvation (see Fig 1K). While contradictory to the original dogma that glucose starvation activates AMPK to upregulate autophagy, our data is consistent with recent data that suggest a more complex relationship between AMPK and autophagy [34,36]. Specifically, it has been postulated that while autophagy may be induced by mild energetic stressors, it may be inhibited by severe stressors, such as complete glucose starvation (as used here), to avoid the energy expenditure required for autophagy and to conserve autophagy resources and protect critical cellular components [37,40,50,51]. Our data demonstrate that aa-starvation triggered robust dephosphorylation of ULK1 at two major inhibitory sites (S758 and S556), while phosphorylation of these sites remained high with glucose starvation. Thus, it is possible that differential phosphoregulation of ULK1, the critical gatekeeper to autophagy induction, underlies the drastically different responses to these two nutrient stressors in our study.

By using carefully defined medias, we established an $EC_{50}$ of approximately 6–7% amino acids towards MTORC1-mediated ULK1 regulation (pULK1-S758) and autophagy vesicle synthesis during a 6 hour cell culture treatment. This demonstrates that the amino acid concentrations found in typical cell culture media (like RPMI-1640, the basis for our medias) are in far excess of a threshold that would induce autophagy, consistent with the original intent of these medias to support cell viability and proliferation for several days in culture [52].

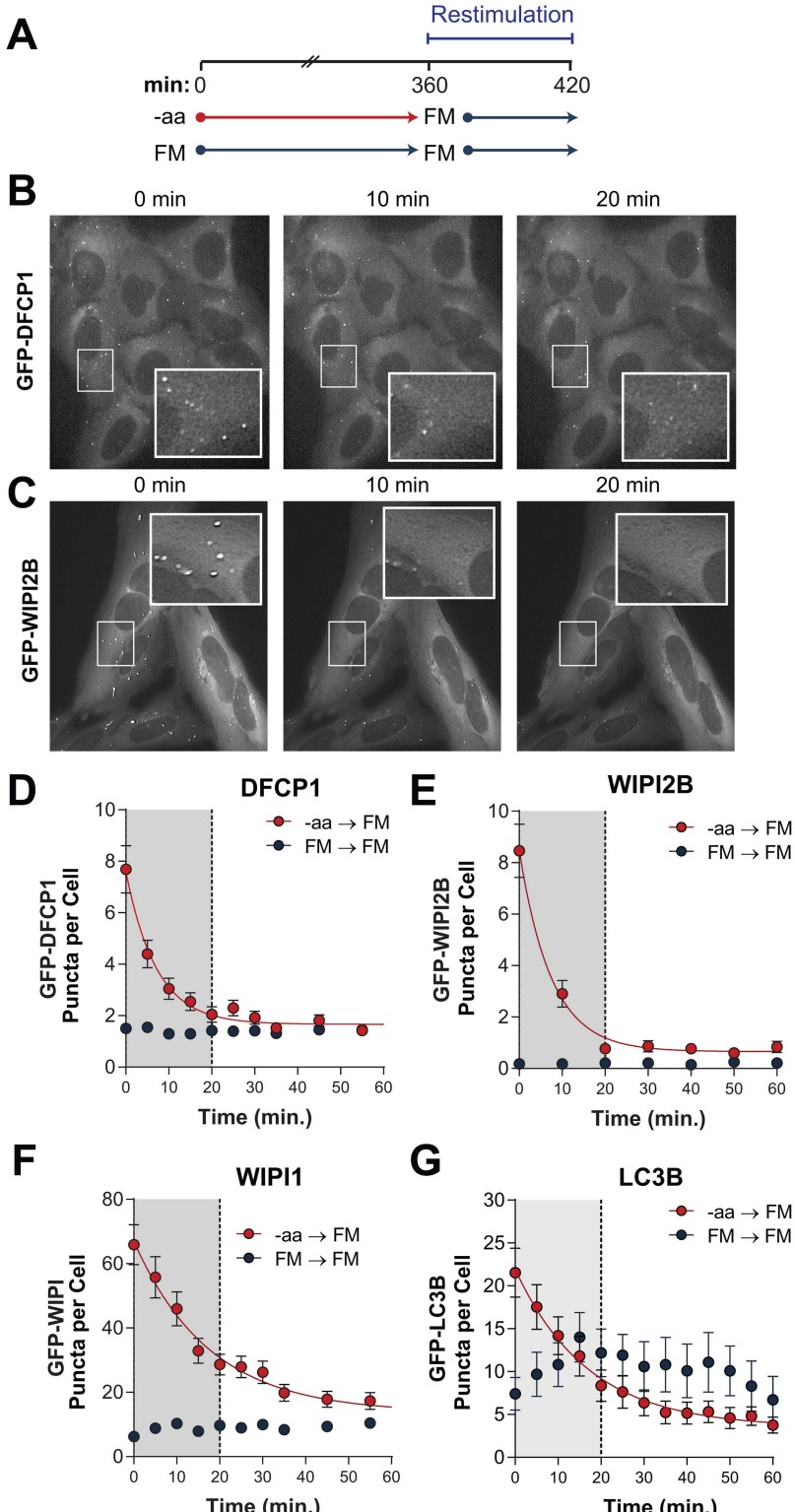

**Fig 5. Autophagy levels are restored immediately upon amino acid replenishment. (A)** Cells were cultured with or without amino acids for 6 hours prior to a restimulation phase of 60 min with FM (containing amino acids). **(B-C)** Representative images of GFP-DFCP1 (B) or GFP-WIPI2B (C) puncta in U2OS cells that were starved of amino acids for 6 hours and subject to aa-restimulation for 0 min (left), 10 min (middle) or 20 min (right). Insets show

2x magnification of indicated region to highlight disappearance of puncta. **(D-G)** DFCP1 **(D)**, WIPI2B **(E)**, WIPI1 **(F)**, and LC3B **(G)** quantified from cells during the restimulation period following -aa (red) or FM (blue) treatments. Symbols are mean GFP-positive puncta per cell and bars are s.e.m. Solid lines are non-linear regression models (one phase exponential decay). Gray shaded area emphasizes restoration to FM levels within 20 min of aa restimulation.

While we draw conclusions from the population averages of data, we captured autophagy measurements from individual cells using live-cell imaging and analysis. Through this exercise, we uncovered significant cellular heterogeneity, despite each line being derived from a single clone. It is possible that the varied cellular responses we saw may relate to the cell cycle; for example, autophagy is inhibited during mitosis in order to protect nuclear contents during cell division [53]. Performing experiments in synchronized cells could address this possibility in future studies. In addition, it is possible that individual cell responses relate to individual gene and protein expression in single cells, which could be explored by cross-comparing autophagy responses with single-cell RNAseq. A limitation of our study was that all markers were studied separately in individual cell lines, so it would be informative to express multiple effectors in a single cell line and confirm whether conserved subpopulations of cells exist that display distinct autophagy responses. Finally, it is intriguing to consider that individual cells responding at the level of autophagy on distinct timescales and with varied magnitudes could be beneficial to a cell population as a whole. While autophagy serves as an important cytoprotective mechanism in response to stress, it can also be detrimental if engaged too robustly or for a prolonged period of time; therefore, a diversity of autophagic responses may be favorable to a growing tumor.

We highlighted a clear difference in the response of two closely related PI(3)P effectors, WIPI1 and WIPI2B, to autophagy induction in our cells. We postulate that WIPI1 is recruited immediately to abundant, pre-existing membrane structures, whereas WIPI2B is induced later and exhibits similar dynamics to DFCP1. Our data is consistent with WIPI2B localizing specifically to early autophagic membranes, and functioning closely with DFCP1 [14]. In contrast, WIPI1 has been found to localize to not only early autophagic membranes, but also plasma membrane, nuclear membrane, endoplasmic reticulum (ER), and LAMP1-positive membranes [54]. We found that WIPI1-positive structures were far more abundant than WIPI2B-positive structures and closer to the number of total 2xFYVE-positive puncta, which labels all PI(3)P-positive membrane compartments, including the endolysosomal pathway. Thus, our data would support a model whereby WIPI1 is exquisitely sensitive to autophagy induction but functions at sites of existing and abundant PI(3)P, perhaps endosomal in origin, in contrast to WIPI2B, which may be more specific to the omegasome and is engaged most significantly after sustained starvation.

We observed an apparent boost in autophagy several hours into amino acid starvation, which co-occurred with a transcriptional program involving a diverse collection of differentially expressed genes, including immediate early genes known to be induced as part of the amino acid response (AAR) [48]. In addition, of the 198 autophagy genes detected in our cells, 36 (18%) were upregulated compared with just 7 (4%) downregulated. This supports a model whereby cells faced with persistent starvation increase autophagy machinery to maintain a high level of autophagy.

After several hours of starvation, we found that cells responded swiftly and completely to the restoration of nutrients. In fact, all markers returned to basal levels within 10–20 minutes of amino acid replenishment, regardless of the initial level of autophagy in each cell. This illustrates that cells are primed to return to their basal state of low autophagy. In this respect, autophagy can be viewed as a robust response to stress but one that is carefully regulated to avoid detrimental consequences.

## Methods

### Mammalian cell culture, reagents, and antibodies

The human osteosarcoma cell line U2OS (HTB-96) was purchased from American Type Culture Collection, and cells maintained in RPMI-1640 medium (Gibco, 11-875-119) supplemented with 10% fetal bovine serum (Corning, 35–010-CV)

and cultured at 37°C in a humidified atmosphere containing 5% $CO_2$. Cells were seeded 48 hours before the start of assays. We generated monoclonal U2OS cell lines expressing the following fluorescent plasmids: *1)* ptfLC3B was a gift from Tamotsu Yoshimori (Addgene plasmid #21074; http://n2t.net/addgene:21074; RRID:Addgene_21074) [55]; *2)* pMXs-puro GFP-DFCP1 was a gift from Noboru Mizushima (Addgene plasmid #38269; http://n2t.net/addgene:38269; RRID:Addgene_38269) [56]; *3)* pMXs-IP-EGFP-mATG5 was a gift from Noboru Mizushima (Addgene plasmid #38196; http://n2t.net/addgene:3819; RRID:Addgene_38196) [57]. We obtained two additional monoclonal U2OS cell lines (GFP-WIPI1-U2OS and GFP-WIPI2B-U2OS) as a kind gift from Tassula Proikas-Cezanne [58] and previously published U2OS-EGFP-2xFYVE [59]. Compound 31 (kind gift from Merck) and Bafilomycin A1 (BafA1; AG Scientific, B1183) stock solutions were prepared in dimethyl sulfoxide (DMSO) (Sigma-Aldrich, D2650). Equal concentrations of DMSO was used for control treatments. Primary antibodies used were Total Acetyl-CoA Carboxylase (ACC; Cell Signaling Technology, 3662), Acetyl-CoA Carboxylase phospho-S79 (Cell Signaling Technology, 11818), S6K1 phospho-T389 (Cell Signaling Technology, 9205), ULK1 phospho-S758 (Cell Signaling Technology, 14202), ULK1 phospho-S556 (Cell Signaling Technology, 5869) and β-actin (Cell Signaling Technology, 3700). IRDye infrared fluorescent 680RD secondary goat anti-rabbit (926–68071) and anti-mouse (926–68070) purchased from LI-COR.

### Defined medias

To prepare defined starvation medias, we reconstituted media containing the components of RPMI-1640, including the following reagents: 1XDPBS (Gibco, 14040−133), Phenol Red (Sigma-Aldrich, P3532), HEPES Buffer 1 M (Gibco, 15630−080), RPMI-1640 Vitamin Mix 100X (Sigma-Aldrich, R7256), RPMI-1640 Amino Acids Solution 50x (Sigma-Aldrich, R7131), sodium bicarbonate (Sigma-Aldrich, S5761), reduced Glutathione (Sigma-Aldrich, G6013), 200 mM Glutamine (Gibco, 25030−081), D-Glucose (Fisher Scientific, D16-500), and dialyzed Fetal Bovine Serum (Sigma-Aldrich, F0392). Amino acid starvation was prepared by adding all components except amino acid solution and glucose starvation media was prepared by adding all components except glucose. See S1 Table for full formulations. Medias were prepared, pH adjusted to 7.2–7.4, sterile-filtered, and stored at 4°C.

### Immunoblot analysis

Cell lysates were prepared in ice-cold lysis buffer [40 mM HEPES pH = 7.4, 1 mM ethylenediaminetetraaceticacid (EDTA), 120 mM sodium chloride, 50 mM bis-glycerophosphate, 1.0% Triton-X 100, 1.5 mM sodium orthovanadate, 50 mM sodium fluoride, and protease inhibitor cocktail (Sigma-Aldrich, P8340)]. Lysate samples were clarified by centrifugation for 10 min at 13,400 rpm and 4°C. Total protein concentration was determined using Protein Assay Dye Reagent Concentrate (Bio-Rad, 5000006). Proteins resolved by pre-cast 4–12% NuPage Bis-Tris Plus Midi gels (Invitrogen, WG1403BOX) or 3–8% NuPage Tris-Acetate Midi gels (Invitrogen, WG1603BOX) and electrotransferred onto either nitrocellulose or polyvinylidene difluoride membranes. Membranes blocked with StartingBlock (TBS) Blocking Buffer (Thermo, 37542) and primary antibodies diluted in StartingBlock T20 (TBS) Blocking Buffer (Thermo, 37543) at 4°C. After three 5 min 0.1% Tween20-TBS washes, membranes incubated in secondary antibodies StartingBlock T20 (TBS) Blocking Buffer for 1 hour at room temperature. Protein bands imaged using LI-COR Odyssey Infrared Imaging System and quantified with LI-COR Image Studio Software.

### Fluorescence microscopy and vesicle quantification

Fluorescent cells (2.5 x $10^4$ cells per chamber) were seeded in 4-chamber 35 mm CELLview dishes with glass bottom (Greiner Bio-One, 627870) and allowed to settle for 48 hours before imaging. Thirty hours post-seeding, media in each chamber was replaced with fresh RPMI-1640 with 10%FBS. Forty-eight hours later cells were washed with warmed 1X DPBS (Gibco, 14190144) and given indicated media. In Fig 3, 1 µM of Compound 31 or vehicle control (DMSO) was administered at the time media was changed with amino acid starvation. For imaging, media was supplemented with

100 nM bafilomycin A1 (BafA1; A.G. Scientific) or an equivalent amount of vehicle (DMSO) for U2OS-ptfLC3B cell lines only for 2 hours as depicted in Fig 1H. All other cell lines contained only relevant starvation medias. For restimulation experiments, after media change cells were cultured for 6 hours in their respective starvation media, at the end of 6 hours imaging was paused to change all media to complete nutrient media. After all conditions had complete media imaging was resumed imaging every position every 5 minutes for 55 minutes. Cells were imaged live by maintaining a humid environment at 37°C and 5% $CO_2$ in an environmental chamber fixed around the microscope stage. Five fields of view per chamber were chosen and NIS Elements software (Nikon) set to automatically image each position every 10 min for 2 hours (U2OS-ptfLC3B) or every 20 min for 6 hours (DFCP1, WIPI1, WIPI2B, and ATG5) using perfect focus to maintain the desired focal plane. One image for each point was obtained before the media change and defined as time zero. For restimulation experiments, media was replaced with complete nutrient media after 6 hours of starvation and imaging continued every 5 minutes for 55 minutes. Fields of view were chosen for their inclusion of healthy cells which were adherent and at the periphery of a cluster. Cells were imaged using a 60X oil objective, in the FITC channel, on a Nikon Ti Eclipse fluorescent microscope. Images were segmented into individual cells by defining regions of interest (ROIs). All images were deconvolved, top-hat transformed (peak identification), and thresholded (intensity) using NIS Elements Software to quantify EGFP-LC3-positive, GFP-DFCP1-positive, GFP-WIPI1-positive, GFP-WIPI2B-positive, EGFP-ATG5-positive, or EGFP-2xFYVE-positive puncta per cell [60]. Processed, quality data was obtained from an average of 58 cells per marker per media condition. Single cell data are accessible online (https://doi.org/10.6084/m9.figshare.28797569).

**LC3-positive AV synthesis and degradation calculations**

GFP-LC3B-positive puncta represent LC3-positive autophagic vesicles (AVs). Cells contain a dynamic visible pool of AVs (that is, the number we quantify by fluorescent microscopy, described above) that is influenced by the addition of newly synthesized AVs and loss of degraded AVs as they fuse with the lysosomal compartment. Therefore, at a given timepoint, the number of AVs present in the cell is equal to the number of AVs that were present at an earlier timepoint *plus* the number of AVs that were synthesized between those two timepoints *minus* the number of AVs that were degraded (in the lysosome) between those two timepoints. This can be summarized by the following simple equation:

$$\mathbf{AV}_{t(x+1)} = \mathbf{AV}_{t(x)} + \mathbf{AV(s)} - \mathbf{AV(d)}$$

Whereby
$\mathbf{AV}_{t(x)}$ is the number of AVs quantified at time x
$\mathbf{AV}_{t(x+1)}$ is the number of AVs quantified at time x + 1 (that is, later than x)
$\mathbf{AV(s)}$ is the number of AVs synthesized between time x and time x + 1
$\mathbf{AV(d)}$ is the number of AVs degraded in the lysosome between time x and time x + 1

The presence of BafA1 prevents lysosomal degradation and therefore, when quantifying AVs from BafA1-treated cells, we set AV(d) = 0. In this case, we can rearrange the equation to find that the number of AVs synthesized (AV(s)) equals the number of AVs at a given time *minus* the number of AVs that was present at time 0. To determine the number of degraded AVs, we use cells not treated with BafA1 (which allows undisturbed degradation) and quantify the number of AVs over time. We use these quantified AVs over time in the equation above along with the value calculated for AV(s) to solve for AV(d). The time of BafA1 exposure must be limited to avoid unintended effects and feedback mechanisms so we perform these LC3 measurements using 2-hour intervals with or without BafA1 and the desired media (FM, -aa, or -Glc). We calculate synthesis and degradation over time and display as the cumulative synthesis and degradation from the start of the first imaging period (t0), which is plotted in Fig 1K, and we use propagation of error rules to determine standard deviation after these calculations. Calculations can be found in S2 Table.

## Amino acid measurements

180,000 U2OS cells were seeded per well of 6-well dishes in 2 ml RPMI-1640 + 10% FBS for 24 hours before being washed and incubated with -aa media for 20 min, 1 hour, or 4 hours (a time 0 control was also harvested for comparison of aa levels). Media was aspirated, cells rinsed twice with ice-cold 0.9% sodium chloride (Sigma S8776), dried completely, and snap-frozen on dry ice. To extract intracellular polar metabolites, 1.5 mL of extraction solvent (40% acetonitrile, 40% methanol, 40% water, v/v) was added to frozen plates, plates were scraped, and extracts collected in 1.5mL Eppendorf tubes. Extracts were incubated on ice for 1 hour, then centrifuged at 15000xg for 10 min. 850 μL of supernatant was collected, dried in a vacuum evaporator, and resuspended in 50uL of water for LCMS analysis. An additional aliquot of 50 μL from each supernatant was pooled, dried, and resuspended and analyzed between every six experimental replicates to monitor instrument and sample drift during analysis. Metabolomics was performed using anion-paired chromatography on an Agilent 1290 UHPLC coupled to an Agilent 6470 QQQ mass spectrometer operated in ESI- in dMRM mode as described previously [61–64]. Peak picking and integration were conducted in MassHunter (v8.0, Agilent). Complete instrument parameters, compound transition list, and peak integration notes are available in S6 Table. Pool sizes of the indicated amino acids were normalized to the abundance in time 0 (non-starved) samples and plotted. All data was captured from 3 biological replicates per condition.

## RNAseq

U2OS cells were plated on 6-well dishes at 180,000 cells per well in 2 ml full media (FM) and incubated at 37°C for 24 hours. The next day, the plating media was aspirated, washed, and replaced with either FM or amino acid-starvation media. Cells were incubated for 1, 2, 4, or 6 hours at 37°C before cells lysed and RNA harvested using Qiagen QIAShredders and RNeasy spin columns following manufacturer's protocols. RNA was quantified using a Qubit fluorimeter and analyzed for integrity using a Bioanalyzer. Total RNA was subject to polyA enrichment and sequenced with 2x50 bp reads on an Illumina NovaSeq (30M raw reads per library). For analysis, quality trimming and adapter removal was performed using Trim Galore, reads were mapped to the human reference genome using STAR, and quality control of trimming and alignment performed with MultiQC. We generated gene counts using STAR and imported into R and analyzed with an internal RNA-seq analysis pipeline (Van Andel Research Institute Bioinformatics and Biostatistics Core).

## Pathway enrichment analysis

Pathway enrichment analysis was performed using Metascape [65] for differentially expressed genes (DEGs) increased during aa-starvation as compared to full media (FM) by a log2-transformed fold-change (log2FC) ≥ 4. We searched for enrichment from a background gene list of 15,869 genes detected in our RNAseq experiment (15,862 recognized by Metascape). We searched all available pathways in Metascape (GO biological processes, Canonical pathways, Hallmark gene sets, Reactome gene sets, WikiPathways, BioCarta gene sets, KEGG pathways, PANTHER pathways). Significantly enriched pathways are identified in S4 Table for 4h and 6h aa-starvation. There were no genes increased by log2FC ≥ 4 at 1h or 2h aa-starvation. Pathways with a p-value < 0.01, a minimum count of 3 genes, and an enrichment factor > 1.5 (ratio between the genes present in the DEG list and the number of genes expected by chance) were grouped into clusters based on similarity (the most statistically significant term within a cluster is chosen to represent the cluster). p-values were calculated based on the cumulative hypergeometric distribution and q-values calculated using the Benjamini-Hochberg procedure to account for multiple testing.

## Statistical analysis

All statistical tests performed in GraphPad Prism. Biological replicates and error bars represent standard error of the mean (SEM), unless otherwise indicated. Related to Fig 4, Fisher's exact test was performed by building a contingency

table with 36 autophagy database genes and 1,606 non-autophagy database genes increased ≥1 log2 FC by 6 hours in -aa versus FM, and 162 autophagy database genes and 14,065 non-autophagy database genes not increased (two-tailed T-test, p = 0.0009). For Fig 3, one-way ANOVA performed with Sidak's multiple comparison test; vehicle versus compound 31 compared at each timepoint.

## Supporting information

**S1 Fig. Raw immunoblot images.** This file contains unmodified raw images of immunoblots included in this study.
(PDF)

**S2 Fig. ULK1 is differentially phosphorylated during amino acid and glucose starvation.** (A-C) U2OS cells were treated with full media (FM), aa-starvation media (-aa), or glucose-free media (-Glc) for the durations indicated before cells lysed and proteins analyzed by immunoblotting. A representative immunoblot is shown in (A). pULK1-S758 (relative to beta-actin) (B) or pULK1-S556 (relative to beta-actin) (C) were normalized to the first timepoint within each treatment and plotted with time (blue: FM; red: -aa; green: -Glc). Symbols are means of triplicate experiments and bars are s.e.m.
(TIF)

**S3 Fig. Amino acid replenishment slows LC3-vesicle synthesis.** (A) U2OS cells expressing GFP-LC3B were cultured for 6 hours in FM or -aa were then stimulated with fresh FM for 3 hours (green: -aa then FM; blue: FM then FM). Bafilo-mycin (BafA1) was supplemented (solid symbols) or not (open symbols) in the replenished FM and GFP-LC3 vesicles per cell were quantified with time. Symbols represent mean and bars represent s.e.m. Data shown at 10 minute intervals (all data available in Table S5). (B-C) GFP-LC3 puncta synthesized (closed symbols) or degraded (open symbols) was quantified during the FM replenishment period and stitched onto data showing hours 4–6 or FM vs -aa treatment (repurposed from Figure 1K). The dashed line indicates time of FM replenishment. The inset in (C) shows magnification of the boxed region in (B).
(EPS)

**S1 Table. Defined media formulations.** Formulation details for custom defined medias used in this study.
(XLSX)

**S2 Table: GFP-LC3 puncta synthesis and degradation in treatment with full media, aa-starvation, and glucose-starvation.** Average GFP-LC3 puncta per cell for cells treated with FM (tab1), amino acid starvation (tab 2), or glucose starvation (tab 3) with or without BafA1. Data captured in 2-hour intervals (1st, 2nd, 3rd datasets indicated). Synthesis and degradation quantified as described in Methods (and SD and SEM calculated using propagation of error rules). The cumulative (stitched) data from time 0 combines the synthesis and degradation values from all 3 datasets into one continuous plot.
(XLSX)

**S3 Table. RNAseq data from cells treated with full media and aa-starvation with time.** Values in C through J are log2-transformed counts per million (CPM); values in K are log2-transformed fold-change in -aa vs FM at 6 hours; in column M, an entry of "yes" indicates gene is described in the Human Autophagy Database (http://autophagy.lu/clustering/; accessed 11/29/2023).
(XLSX)

**S4 Table. Pathway enrichment analysis for genes increased during aa-starvation.** Pathways enriched among genes changing in expression during 6h (A) or 4h (B) aa-starvation. Count is number of genes from DEG input list present in the pathway; % is percentage of genes in the DEG list that are found in the pathway.
(XLSX)

**S5 Table. GFP-LC3 synthesis and degradation in cells replenished with full media following aa-starvation.** Average GFP-LC3 puncta per cell for cells treated with FM 6h and replenished with fresh FM for 120 min (columns A-O) or amino acid starved 6h then replenished with FM (columns Q-AE). Synthesis and degradation quantified as described in Methods (and SD and SEM calculated using propagation of error rules).
(XLSX)

**S6 Table. Mass-spectrometry technical file.** This file contains technical details relating to the amino acid measurements made by mass-spectrometry, including ion paired transition list (tab 1), ion paired MS parameters (tab 2), ion paired LC parameters (tab 3), and integration notes (tab 4).
(XLSX)

## Acknowledgments

We thank members of the MacKeigan laboratory for their critical discussions and feedback. We thank the Van Andel Institute Cores for providing mass spectrometry, genomics, and bioinformatics facilities and services. We also thank William Hlavacek, Yen Ting, and Song Feng of the Los Alamos National Laboratory for their critical discussions and support for this work.

## Author contributions

**Conceptualization:** Katie R. Martin, Stephanie L. Celano, Jeffrey P. MacKeigan.

**Funding acquisition:** Katie R. Martin, Russell G. Jones, Jeffrey P. MacKeigan.

**Investigation:** Katie R. Martin, Stephanie L. Celano, Jessica D. Guillaume, Ryan D. Sheldon.

**Methodology:** Katie R. Martin, Stephanie L. Celano, Jessica D. Guillaume, Ryan D. Sheldon.

**Project administration:** Stephanie L. Celano, Jeffrey P. MacKeigan.

**Resources:** Russell G. Jones, Jeffrey P. MacKeigan.

**Supervision:** Katie R. Martin, Russell G. Jones, Jeffrey P. MacKeigan.

**Validation:** Stephanie L. Celano, Jessica D. Guillaume.

**Visualization:** Katie R. Martin, Stephanie L. Celano, Jessica D. Guillaume, Ryan D. Sheldon, Jeffrey P. MacKeigan.

**Writing – original draft:** Katie R. Martin, Stephanie L. Celano.

**Writing – review & editing:** Katie R. Martin, Stephanie L. Celano, Ryan D. Sheldon, Russell G. Jones, Jeffrey P. MacKeigan.

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
