## [Decision Letter · Decision Letter 0]

14 Aug 2025

Thank you for submitting your manuscript to PLOS ONE. After careful consideration, we feel that it has merit but does not fully meet PLOS ONE’s publication criteria as it currently stands. Therefore, we invite you to submit a revised version of the manuscript that addresses the points raised during the review process.

We look forward to receiving your revised manuscript.

Kind regards,

Mohamed Abdelkarim

Academic Editor

PLOS ONE

Journal Requirements: 

3. Thank you for uploading your study's underlying data set. Unfortunately, the repository you have noted in your Data Availability statement does not qualify as an acceptable data repository according to PLOS's standards.

Reviewers' comments:

Reviewer's Responses to Questions

**Comments to the Author**

1. Is the manuscript technically sound, and do the data support the conclusions?

Reviewer #1: Yes

Reviewer #2: Yes

2. Has the statistical analysis been performed appropriately and rigorously?

Reviewer #1: Yes

Reviewer #2: Yes

3. Have the authors made all data underlying the findings in their manuscript fully available?

Reviewer #1: Yes

Reviewer #2: Yes

4. Is the manuscript presented in an intelligible fashion and written in standard English?

Reviewer #1: Yes

Reviewer #2: Yes

Reviewer #1: This study quantitatively measures autophagy components such as LC3, DFCP1, WIPI1, and WIPI2B under amino acid and glucose starvation using live-cell imaging in U2OS cells. The results demonstrate that autophagy is activated under amino acid starvation but not under glucose starvation. Furthermore, the study reveals transcriptional upregulation of autophagy-related genes that help sustain elevated autophagy levels over an extended period under amino acid starvation. The authors also show that reintroducing amino acids to starved cells restores autophagy to its basal state. However, major work is needed to clearly highlight the novelty of the study and how it addresses current gaps in knowledge.

Major comments:

Novelty and rationale of the study:

The current introduction would benefit from a more clearly articulated rationale that highlights both the specific limitations of prior studies and how this work would address them. While the interplay between glucose, amino acids, and autophagy has been extensively explored and recognized as context-dependent, the manuscript could better clarify the specific mechanistic insights it aims to contribute.

A potentially novel aspect of this study is its kinetic profiling of autophagy components, which can be utilized for modeling autophagy dynamics—an area less explored quantitatively. Refocusing the manuscript around this dataset and its relevance to predictive modeling could significantly strengthen its conceptual impact of the work.

The potential reasons for the lack of autophagy activation under glucose starvation remain unexplored. This could be studied by comparing the phosphorylation states of ULK1 at various sites during amino acid versus glucose starvation. Additionally, analyzing the effects of combined amino acid and glucose starvation may reveal important insights into the dynamics of mTORC1, AMPK, and ULK1 signaling.

Fig 1H-K experiment is well designed to calculate autophagy flux- please refer to these papers to calculate the autophagy flux for all three conditions. - . https://doi.org/10.1080/15548627.2022.2117515, 10.4161/15548627.2014.973338. This would assist the comparison of autophagy flux rates in more straightforward and interpretable manner.

As noted in the discussion, glucose deprivation appears to reduce autophagy flux—quantifying this would support the observation (See point previous point).

Consider pathway enrichment analysis of differentially expressed genes at various timepoints during amino acid starvation to better understand involved pathways.

Figures 2 and 5D–G use GFP sum intensity, while Figure 1 uses GFP puncta to assess autophagy activity—please clarify this discrepancy in measurement.

The authors raise an interesting point regarding the disassembly or degradation of autophagy vesicles following amino acid replenishment. This phenomenon could be further explored by monitoring autolysosomes or assessing LC3–LAMP1 colocalization under bafA1 treatment. This would provide novel insights into the change in autophagy flux and state of autophagy vesicles after replenishment.

Minor comments:

Please rework the abstract wording- “quantitative catalog” . Catalog suggests a comprehensive list of cell lines tested under different media conditions.

Use densitometry to quantify western blot results quantitatively.

Consider consolidating Figures 2A–2D into a single composite plot to facilitate direct comparison across conditions. Uniform axes and color-coding would enhance interpretability.

Clarification of Sigmoidal Fit (Figure 2B): The current legend refers to a sigmoidal dose-response fit yielding an EC50 of 6.4% aa (± 1.2% aa), but 2B depicts only the Relative pULK1-S758 value at 10% aa. Please address this and explicitly state the timepoint used for curve fitting in both the legend and the main text.

Include a scale bar in figure 3D.

Reviewer #2: 1. The manuscript reports that glucose starvation does not induce autophagy in U2OS cells, which contrasts with some existing literature. The authors should discuss potential reasons for this discrepancy.

2. The study notes significant cellular heterogeneity in autophagy responses, which is intriguing. But, the underlying causes are not explored. The authors could strengthen the discussion by proposing experiments or hypotheses to explain this variability.

3. The authors should discuss potential signaling pathways and cite relevant literature to contextualize their findings.

4. The use of BafA1 to measure LC3 accumulation is well-justified, but the manuscript could clarify whether the observed differences in LC3 puncta reflect changes in autophagosome synthesis, degradation, or both.

5. The study focuses on U2OS cells, but autophagy responses may vary across cell types. The authors should discuss whether their findings are likely to be generalizable or cell line-specific.

**Do you want your identity to be public for this peer review?** For information about this choice, including consent withdrawal, please see our Privacy Policy

Reviewer #1: No

Reviewer #2: **Yes: ** Korrapati Narasimhulu

---

## [Author Response · Author response to Decision Letter 1]

2 Dec 2025

Manuscript ID #PONE-D-25-35473

Response to Reviewers

We appreciate the time spent on this manuscript by our reviewers, as well as the constructive suggestions they have made to improve the significance and quality of our work. Answers to each comment presented by the reviewers are detailed below. We feel that the changes we have made in response to these comments strengthen the data that was presented before, and provide additional support for our conclusions.

Referee 1: The current introduction would benefit from a more clearly articulated rationale that highlights both the specific limitations of prior studies and how this work would address them. While the interplay between glucose, amino acids, and autophagy has been extensively explored and recognized as context-dependent, the manuscript could better clarify the specific mechanistic insights it aims to contribute. A potentially novel aspect of this study is its kinetic profiling of autophagy components, which can be utilized for modeling autophagy dynamics—an area less explored quantitatively. Refocusing the manuscript around this dataset and its relevance to predictive modeling could significantly strengthen its conceptual impact of the work.

We are grateful to the reviewer for this constructive feedback. We have expanded our Introduction to include more information on our rationale, including the evolving knowledge surrounding glucose, AMPK, and autophagy (see revised text on page 4 of Introduction) as well as the value of our data being captured with well-controlled media inputs and at high temporal resolution, addressing a limitation of prior studies that has contributed to ambiguity in this research area (see revised text on pages 5-6 of Introduction).

Referee 1: “The potential reasons for the lack of autophagy activation under glucose starvation remain unexplored. This could be studied by comparing the phosphorylation states of ULK1 at various sites during amino acid versus glucose starvation. Additionally, analyzing the effects of combined amino acid and glucose starvation may reveal important insights into the dynamics of mTORC1, AMPK, and ULK1 signaling.”

We appreciate this suggestion. To address this, we used immunoblotting to quantify two critical phosphorylation sites of ULK1, pSer758 and pSer556, in U2OS cells with FM, -aa, or -Glc from 15 minutes to 6 hours. We found that while aa-starvation triggered dephosphorylation of both sites, consistent with reported global dephosphorylation of ULK1 with aa-starvation, glucose starvation maintained high and sustained levels of both sites. The failure to relieve the inhibitory phosphorylation at Ser758 and high levels of pSer556, which has recently been found to associate with reduced ULK1 activity and autophagic flux, suggests that while glucose starvation indeed activates AMPK, the phosphorylation profile of ULK1 favors inhibition rather activation. This may relate to the fact that it has been reported that while mild energetic stress may activate AMPK to support autophagy, severe stress, such as the complete glucose starvation we tested, activates AMPK in such a way that it represses ULK1 and autophagy. These new data can be found in new Figure S1 and revised text on page 8-9 of Results.

Referee 1: “Fig 1H-K experiment is well designed to calculate autophagy flux- please refer to these papers to calculate the autophagy flux for all three conditions. - . https://doi.org/10.1080/15548627.2022.2117515, 10.4161/15548627.2014.973338. This would assist the comparison of autophagy flux rates in more straightforward and interpretable manner… As noted in the discussion, glucose deprivation appears to reduce autophagy flux—quantifying this would support the observation (See point previous point).”

Referee 2: “The use of BafA1 to measure LC3 accumulation is well-justified, but the manuscript could clarify whether the observed differences in LC3 puncta reflect changes in autophagosome synthesis, degradation, or both.”

We appreciate this constructive feedback from both reviewers and feel addressing it has now significantly improved our manuscript. We have now calculated LC3 synthesis and degradation as indicators of autophagy flux leveraging vesicle quantification from each media with and without BafA1 treatment, using a straightforward model similar to that described in Loos et al., 2014 (1). Please see revised Figure 1K, description in the text on page 8 of Results, and new section in Methods called “LC3-positive Synthesis and Degradation Calculations” on pages 20-21. Moreover, we provide the data as tables in new Table S2. Importantly, this allows us to firm demonstrate that autophagic vesicle synthesis (and concomitant degradation) are indeed suppressed by glucose starvation, while being significantly upregulated during aa-starvation.

Referee 1: “Consider pathway enrichment analysis of differentially expressed genes at various timepoints during amino acid starvation to better understand involved pathways.”

We performed pathway analysis using Metascape on genes increased by aa-starvation compared to full media (log2FC > 4) or). See text on pages 12-13 of Results and new Table S4. We identified pathways relating to stress responses, including “NGF-stimulated transcription” and “Response of EIF2AK1 (HRI) to heme deficiency” which involve the integrated stress response and immediate early gene induction, among upregulated genes during aa-starvation. This is consistent with our observation that many immediate early genes were increased along with autophagy machinery. It should be noted that we did not identify significant pathway enrichment among downregulated genes using this same approach so we did not include that analysis in the revised manuscript, instead focusing on the enrichment of pathways among upregulated genes.

Referee 1: “Figures 2 and 5D–G use GFP sum intensity, while Figure 1 uses GFP puncta to assess autophagy activity—please clarify this discrepancy in measurement.”

We appreciate the reviewer noting this discrepancy. At times, our lab has found that sum intensity, which is the sum of fluorescent intensity units of all pixels in puncta, is useful when measuring GFP-LC3-positive puncta because it can circumvent errors introduced when vesicle abundance is very high (e.g., with starvation plus BafA1). However, upon reevaluation of the data presented in this paper, we have determined that in most cases, we could display data as puncta (or vesicles) per cell in order to aid the interpretability of the results and make comparisons of the various markers clearer. Please find updated units in data presented in revised Figure 1K, revised Figure 5D-G, and new Figure S2. Despite this, we maintained the use of sum intensity units in Figure 2 because this figure focused on late stage aa-starvation (4-6 hours) and from cells treated with BafA1 where LC3 vesicles were most abundant. Moreover, we derived rates of LC3 vesicle synthesis in this figure by linear regression; therefore, it was imperative to use kinetic data in the linear range. We feel the data is most rigorous and interpretable within this Figure if the data is analyzed on a sum intensity scale. For your understanding, we provide two panels below comparing GFP-LC3 puncta per cell (left) and GFP-LC3 sum intensity per cell (right) at hours 4-6 of aa-starvation in the presence of BafA1 captured as part of the aa-titration data for Figure 2. Note that the abundance of puncta per cell approaches saturation at the final timepoints (plateaus) whereas the sum intensity of these puncta maintains linearity.

Referee 1: “The authors raise an interesting point regarding the disassembly or degradation of autophagy vesicles following amino acid replenishment. This phenomenon could be further explored by monitoring autolysosomes or assessing LC3–LAMP1 colocalization under bafA1 treatment. This would provide novel insights into the change in autophagy flux and state of autophagy vesicles after replenishment.”

Thank you for this feedback. To explore this, we quantified LC3 vesicle dynamics during the nutrient restimulation period in the presence and absence of BafA1. We reasoned that because BafA1 prevents lysosome degradation, if the vesicles that disappeared quickly in restimulation were doing so from lysosomal degradation, the decrease should be prevented with the addition of BafA1. Indeed, this is what we observed (see new Figure S2A). Moreover, we expanded our calculations of AV synthesis and degradation to include this restimulation dataset and observed that upon nutrient replenishment, vesicle synthesis seems to slow immediately to a rate similar to that of nutrient-replenished previously fed cells (see the slope of the solid red and solid blue symbols after 360 minutes in new Figure S2B). In contrast, AV lysosomal degradation continued early in the restimulation period at a rate similar to the previous rate during the aa-starvation before leveling out to balance with synthesis by about 30 minutes after restimulation (new Figure S2C). We have also provided these calculations in new Table S5. Please also refer to revised text on pages 13-14 of Results.

Referee 1: “Please rework the abstract wording- “quantitative catalog” . Catalog suggests a comprehensive list of cell lines tested under different media conditions.”

This was a helpful interpretation of the statement as originally written, which we have now updated to read more accurately “quantitative dataset” (see updated text on page 2 of the Abstract).

Referee 1: “Use densitometry to quantify western blot results quantitatively.”

We appreciate this suggestion. The immunoblots were captured using an Odyssey CLX machine and near-infrared fluorescence imaging, which are ideal for quantification. We quantified the pACC-S79 and pS6K1-T389 immunoblots displayed in Figure 1B and 1C – please see quantification in new Figure 1D and 1E). Moreover, we also quantified the newly generated immunoblots for ULK1-pS758 and ULK1-pS556, which we provide in new Figure S1B and S1C.

Referee 1: “Consider consolidating Figures 2A–2D into a single composite plot to facilitate direct comparison across conditions. Uniform axes and color-coding would enhance interpretability…Clarification of Sigmoidal Fit (Figure 2B): The current legend refers to a sigmoidal dose-response fit yielding an EC50 of 6.4% aa (± 1.2% aa), but 2B depicts only the Relative pULK1-S758 value at 10% aa. Please address this and explicitly state the timepoint used for curve fitting in both the legend and the main text.”

Thank you for your thorough review of Figure 2 and its legend. First, we have focused on the final period of starvation to improve clarity and interpretation of the data presented in this figure. Revised Figure 2A now displays the relative level of ULK1-pS758 after 6 hour treatment across the different aa concentrations in a single panel to aid the comparison. We have also matched the color-coding across all panels. The sigmoidal fits referred to in the text were inadvertently omitted from the version of Figure 2 uploaded originally; we sincerely appreciate this referee catching this in their thorough review. The sigmoidal curve and EC50 determinations were made for both pULK1-S758 and GFP-LC3 synthesis across the different aa concentrations, which are now included in revised Figure 2B and 2D. We used the pULK1-S758 quantification at 6 hours of treatment and the rate of GFP-LC3 synthesis from 4-6 hours (which is independently displayed in revised Figure 2C) and intended to highlight the similarity in influence of aa depletion of both of these markers of autophagy. We hope you find this newly revised Figure 2 and legend to be coherent and informative.

Referee 1: “Include a scale bar in figure 3D.”

Please see revised Figure 3D, which now includes scale bars in all panels. We also want to point out that Figure 3E is presented in a slightly different manner (a bar chart comparing an early and late point in aa-starvation, instead of a complete time series). We feel this display allows for the effects of compound 31 to be best represented (the original legend for Figure 3 was accurate for this bar chart).

Referee 2: ”The manuscript reports that glucose starvation does not induce autophagy in U2OS cells, which contrasts with some existing literature. The authors should discuss potential reasons for this discrepancy.”

The published literature concerning the role of AMPK and autophagy during energetic stress is indeed conflicting, with recent published commentaries discussing this at length, which we now cite (2, 3). We now explain and discuss this in more detail – please see revised text on pages 4-5 of the Introduction. Our new analysis of ULK1 phosphorylation demonstrates very different states between amino acid and glucose starvations, which may explain the failure to induce autophagy during glucose starvation in our experiments (please see new Figure S1). Moreover, it has been suggested that AMPK indeed activates autophagy during mild or moderate energetic stress but during a severe crisis, such as the complete deprivation of glucose we subjected our cells to, AMPK may actually inhibit autophagy in an effort to preserve the cell. We feel that by using precisely-formulated medias identical with the exception only of aa or glucose, quantifying markers that span multiple stages of autophagy, and capturing data with the comprehensive temporal resolution (from 5 minutes through 6 hours), our data is sound to be able to conclude with confidence that glucose starvation fails to induce autophagy under the reported conditions.

Referee 2: “The study notes significant cellular heterogeneity in autophagy responses, which is intriguing. But, the underlying causes are not explored. The authors could strengthen the discussion by proposing experiments or hypotheses to explain this variability.”

Thank you for this feedback. We have now added to the discussion of this observation – please see revised text on page 16 of the Discussion.

Referee 2: “The authors should discuss potential signaling pathways and cite relevant literature to contextualize their findings.”

Thank you for this feedback. We discuss the regulation of autophagy by MTOR and AMPK signaling pathways on page 4 of Introduction and a new pathway enrichment analysis in revised text on page 12-13 of Results.

Referee 2: “The study focuses on U2OS cells, but autophagy responses may vary across cell types. The authors should discuss whether their findings are likely to be generalizable or cell line-specific.”

We chose to complete our analyses in the U2OS cells, given evidence of robust autophagy and MTOR/AMPK signaling in this cell line. U2OS cells are a widely used and well-studied cell model for mTOR signaling. Research shows that the mTOR pathway is highly active in these cells and is a key driver of their growth and survival, and importantly, the autophagy pathway is highly responsive, dynamic, and intact. Because the autophagy pathway and the machinery explored are highly conserved, we expect our findings not to be cell line specific and could translate well to other cancer cell lines as well. We added a statement concerning this – see revised text on page 15 of the Discussion.

REFERENCES

1. Loos B, du Toit A, Hofmeyr JH. Defining and measuring autophagosome flux-concept and reality. Autophagy. 2014;10(11):2087-96.

2. Mandic M, Misirkic Marjanovic M, Janjetovic K, Bosnjak M, Harhaji-Trajkovic L, Trajkovic V, Vucicevic L. Multifaceted role of AMPK in autophagy: more than a simple trigger? Am J Physiol Cell Physiol. 2025;329(5):C1380-C97.

3. Kim DH. Contrasting views on the role of AMPK in autophagy. Bioessays. 2024;46(3):e2300211.

---

## [Decision Letter · Decision Letter 1]

30 Dec 2025

Quantitative and Temporal Analysis of Autophagy: Differential Response to Amino Acid and Glucose Starvation

PONE-D-25-35473R1

Dear Dr. Jzffrey,

We’re pleased to inform you that your manuscript has been judged scientifically suitable for publication and will be formally accepted for publication once it meets all outstanding technical requirements.

Kind regards,

Mohamed Abdelkarim

Academic Editor

PLOS One

Additional Editor Comments (optional):

Reviewers' comments:

Reviewer's Responses to Questions

**Comments to the Author**

Reviewer #3: All comments have been addressed

2. Is the manuscript technically sound, and do the data support the conclusions?

Reviewer #3: Yes

3. Has the statistical analysis been performed appropriately and rigorously?

Reviewer #3: Yes

4. Have the authors made all data underlying the findings in their manuscript fully available?

Reviewer #3: Yes

5. Is the manuscript presented in an intelligible fashion and written in standard English?

Reviewer #3: Yes

Reviewer #3: Neutral & clear

The authors have adequately addressed the reviewers’ comments, and the revised manuscript is satisfactory. I therefore recommend acceptance of the paper.

Slightly more formal (typical journal style)

The authors’ responses to the reviewers’ comments are appropriate and satisfactory. The revisions have improved the manuscript, and I recommend acceptance of the paper.

Formal & academic

The authors have responded appropriately to all comments raised during the review process. The revised version of the manuscript is satisfactory, and I recommend its acceptance for publication.

Very concise (editor-friendly)

The authors’ response is satisfactory. I recommend acceptance of the manuscript.

Positive and encouraging

The authors have carefully addressed all comments, and the revised manuscript meets the required standards. I recommend acceptance of the paper.

**Do you want your identity to be public for this peer review?** For information about this choice, including consent withdrawal, please see our Privacy Policy

Reviewer #3: **Yes: ** Mohamed Abdelkarim

---

## [Editor Report · Acceptance letter]

PONE-D-25-35473R1

PLOS One

Dear Dr. MacKeigan,

I'm pleased to inform you that your manuscript has been deemed suitable for publication in PLOS One. Congratulations! Your manuscript is now being handed over to our production team.

Kind regards,

on behalf of

Dr. Mohamed Abdelkarim

Academic Editor

PLOS One